# Improving Energy Management through Demand Response Programs for Low-Rise University Buildings

**Akeratana Noppakant and Boonyang Plangklang \***

Department of Electrical Engineering, Faculty of Engineering, Rajamangala University of Technology Thanyaburi, Pathum Thani 12110, Thailand
* Correspondence: boonyang.p@en.rmutt.ac.th; Tel.: +66-86-899-2996

**Abstract:** Recently, energy costs have increased significantly, and energy savings have become more important, leading to the use of different patterns to align with the characteristics of demand-side load. This paper focused on the energy management of low-rise university buildings, examining the demand response related to air conditioning and lighting by measuring the main parameters and characteristics and collecting and managing the data from these parameters and characteristics. This system seeks to control and communicate with the aim of reducing the amount of peak energy using a digital power meter installed inside the main distribution unit, with an RS-485 communication port connected to a data converter and then displayed on a computer screen. The demand response and time response were managed by power management software and an optimization model control algorithm based on using a split type of air conditioning unit. This unit had the highest energy consumption in the building as it works to provide a comfortable environment based on the temperatures inside and outside the building. There was a renewable energy source that compensated for energy usage to decrease the peak load curve when the demand was highest, mostly during business hours. An external power source providing 20 kWh of solar power was connected to an inverter and feeds power into each phase of the main distribution. This was controlled by an energy power management program using a demand response algorithm. After applying real-time intelligent control demand-side management, the efficient system presented in this research could generate energy savings of 25% based on AC control of the lighting system. A comparison of the key system parameters shows the decrease in power energy due to the use of renewable energy and the room temperature control using a combination of split-type air conditioning.

**Keywords:** building management system; demand response; energy management system; low-rise university building; real-time intelligent control

## 1. Introduction

Today, the issue of energy costs is becoming increasingly important as a result of the global energy crisis. At the same time, the growing use of modern appliances is resulting in ever-increasing energy consumption. Energy consumption management has become increasingly important in all sectors. Therefore, energy savings from existing facilities have become a major priority. However, the traditional approach using fossil fuels and natural gas and energy resources is inappropriate in balancing the energy supply and demand-side management [1]. Inadequate energy consumption control cannot properly respond to the user's energy consumption. The energy usage sector consists of residential, commercial, public, and industrial. The energy consumption of buildings is a crucial factor in energy usage across all sectors. Buildings generate the highest energy consumption and need to be managed through the optimal strategies. Therefore, the demand-side load of buildings is incorporated into many types of equipment such as air conditioning systems, illumination systems, computer systems, transportation systems, plumbing and heating systems, and facilities management equipment [2]. The demand-side management of buildings needs to

be controlled to reduce the energy consumed by users. The choice of renewables now being applied to buildings, generating power by using a photovoltaic system, is presented in [3]. A smart electric power grid and the integration of communications technology can manage the delivery of electricity from a power plant grid to the consumer. The data network enables a two-way flow of information. The energy demand can provide the supply balance in real-time, as it is focused on the demand side and load saving by improving the energy consumption pattern [4,5]. Machine learning is adapted by forecasting data on the energy efficiency of buildings. The results of machine learning are represented in terms of the type and nature of the building and the equipment energy consumed [6]. Therefore, non-intrusive load management (NILM) techniques are combined with a demand response program to reduce the peak demand of the electrical load from the energy demand of the building. This algorithm focuses on minimizing the energy consumption and energy cost [7,8]. The mining of big building operational data for energy efficiency enhancement has been adapted via unsupervised data analysis and can be used to predict the energy data of the modern building, as presented in [9].

For modern energy consumption management systems designed for consumers, an intelligent network system has been developed to communicate and interact with automatic and efficient functioning [10]. Implementing a future smart grid could optimize real-time operation and energy consumption management [11,12]. Sensors installed in the smart grid can detect and display the Internet of Things (IoT) data in real-time for an unlimited period.

The operational effect of these characteristics is to reduce the energy consumption significantly. Representations for commercial buildings include the building type, destination control, and climate zone. This leads to a data-driven analysis of knowledge to analyze base data, adding management strategies and then converting them into rules for management control. The approach adopting the demand response (DR) decreases the electric peak power demand consumption from the load-side demand. The electricity load demand from the peak demand triggers another source of electric energy, where the electric energy efficiency is higher. Energy conservation configurations include systems such as rooftop solar cell electric power, using an electricity storage battery, or others such as power-intensive loads, which use intelligent (smart) equipment technologies. The management of the algorithm to increase the electric power efficiency and decrease the electricity load peak is incorporated into electric power grid distribution networks. To summarize, the demand response using the time response optimization algorithm is the basis of the smart electric power grid. Research papers have been published on the electric power of buildings. These have explored the demand response optimization algorithms for industrial equipment and home appliance load scheduling. These papers are comprehensive, but not adequately focused. The forecasted building electric usage data and patterns were used to set the optimization algorithm objective. Additionally, a technique to prove the methodology's efficiency was included, and the methodology can be used and applied to similar problems.

Most of the research papers in the worldwide database have focused on the utilization of HVAC for energy management systems. According to the behavior of the energy demand in buildings from the analysis of the usage data results, air conditioners consume the most energy, representing 65% of the total energy demand when compared to other building equipment. Typically, the air conditioner characteristics of evaporative coolers in dry areas result in low efficiency [13].

In this paper, the authors focused on the electric grid building demand response and time response optimization algorithm for the split-type air conditioning and lighting systems, analyzing the benefits and challenges regarding the algorithm's design efficiency and its implementation. The electric grid building demand response is needed to present the problem's background, and the different scenarios introduce the time response model to the study problem. A review was conducted to analyze the challenges and relevant results and data for the future demand response and time response optimization algorithms. This work aimed to present the control function of the electric grid for the demand response and to review the methods to obtain different solutions for the electric grid building

demand response. The paper's methodology is represented by matching a building's energy demand requirements from all of its electrical appliances and devices such as the air conditioning and lighting system, etc.

In the remainder of this paper, we attempted to present the details of the BEMS system. Section 2 reviews the previous work on the building demand response to improve the energy saving for split-type air conditioners. Section 3 provides the details of the optimization of the demand response in controlling the air conditioning and lighting system. Section 4 explains the purpose, the implementation of the building demand response (BDR), and the challenges. Finally, we present the results and conclusions.

## 2. Brief Literature Review

Most smart buildings have many characteristics aside from being connected to the electric microgrid including various types of renewable energy systems such as rooftop solar power and others, which can operate with an electricity storage battery. The demand response is one of the new developments in electricity that is intended to engage consumers in improving their electric energy consumption patterns. In Thailand, many types of demand response are already provided through demand response programs.

The building demand response can operate a schedule in real-time via the hours/day; generally, for a building demand response, a time response algorithm is used to optimize the efficiency by decreasing the electricity consumption costs for the consumer, thereby increasing the maximum profit for the end user [14]. A literature review presented real-time data with energy program solutions including the program's algorithm and design [15]. For the intelligent building (BEMS), most appliances were individually connected, with different mechanisms to control each appliance [16]. The essential data accumulated simultaneously from many appliances are collected into big data. Big data analytics can improve the energy usage of consumer behavior and manage and set up the real-time price design and performance of the demand response program [17].

Regarding the electric grid building demand response, a time response algorithm seeks to set a comfort zone operational level as a priority and create an operational pattern for the control of the air conditioning system and lighting appliances [18,19].

The calculation of the electric load and the algorithm configuration depend on the categories of the electricity load type, the time of electricity use, the equipment or appliance service maintenance schedule, and other technical constraints related to the available renewable energy generation electricity capacity [20].

Figure 1 shows the details of the general demand response; these were separated into two categories, which include quantity-based programs and price-based programs. The quantity-based programs include the option of controlling demand. The price-based programs include an external source option to confirm the buyer decision, choose the electricity cost, and determine the time of use to purchase electricity.

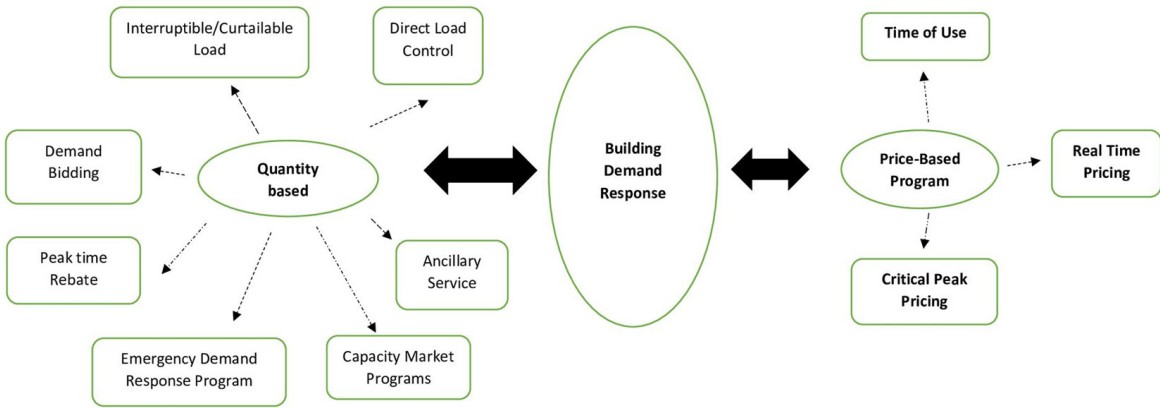

**Figure 1.** Classification of demand response.

The demand response, based on consumers changing their consumption behavior, is categorized into two groups: one is price-based, and the other is intensive-based. Each one has several sub-groups [21].

The BEMS performance is related to the amount of energy consumed [15]. Companies provide electricity savings by tracking and monitoring the users' electricity usage patterns. However, the demand response control mechanisms have undesirable operational times including those for air conditioning systems, lighting, and others.

## 3. Optimization of Demand Response

The combination of BEMS includes an integer and non-linear programming model for dynamic characteristics. The building model description, for this case, is related to hours or days ahead of a schedule, monitoring in real-time the system condition. The building model is constrained by the load type, pricing scheme, decision variables involved, and the capacity for time response communication and control in many solutions. Finally, one can determine the minimum requirement in demand response management programming.

### 3.1. Methodology

The approach to this topic was based on the published demand response papers and journals that identified the challenges of studying the building demand response and time response optimization models.

The first consideration was based on the average hourly electric power demand and the load required to identify the use of a power generator and a renewable electricity power source, alternating with the use of the electric grid over a specific time. The second consideration was based on mathematical models and strategy simulation models; however, some studies have considered more realistic problems and focused on the future scope of research. The third consideration was based on algorithm models and solution factors such as electricity load variations, usage patterns, and other infrastructure involved, in a unified approach to fit most of the building demand response problems.

The objectives of the demand response for the BEMS level were related by including a minimum production cost, maximum comfort, decreased load peak, and self-consumption via renewable energy generation, aggregated with the data of the minimum cost and maximum self-comfort in a social situation. It is constrained by the BEMS level of thermostatic and non-thermostatic control (air conditioning system), lighting and ventilation control, parameters, time usage schedule regarding the collection of data from the air conditioning system control, electricity cost, and the variable electricity load demand. The algorithm setting for process control was determined by the benefits and limitations of the use of multiple choices of renewable energy sources. The data from the electric grid and available renewable energy were used during planning to schedule for the best use between the building and the electric grid. The optimal solution of the problem is needed to ensure that it is suitable for many programs in BEMS systems with wireless, PLC, and SCADA technology, depending on the hierarchical structure of the building and electric grid.

The building demand response types have many factors, creating limitations for each characteristic such as building electrical power control management. The constraints include the limitations of the main appliances consuming electricity and communicating with the electric grid by the demand response and time response algorithm. The benefits include the multiple choices of the available renewable energy to compensate for the main power in peak periods. Finally, it is necessary to consider the communication components using wired and wireless technology such as PLCs and SCADA.

Most research papers have been presented using different techniques to control the demand response with the time response algorithms for BEMS. The BEMS are focused on power generation, using additional free energy from renewable energy generation models. Therefore, the improvement seeks to decrease the energy need; this is one such case affected by the challenges of implementing smart grid technology.

The structured design solution from the data results from the analysis process performed as part of the energy efficiency evaluation must also be considered.

- The renewable energy source is integrated into the building.
- Each floor is evaluated for the determination of the efficiency of the air conditioning system.
- The determination of the air conditioning, lighting, and appliance load can be managed.
- The relevant data on non-renewable energy in the preparation methodology of the performance management system are analyzed.

According to previous researchers, the energy performance does not reflect the significant problem wherein the data are imported from the parameters of the energy performance design, with non-renewable primary energy. In many cases, this presents an important problem in maintaining the energy level as the EP value is significant to the primary process.

The EP value is shown in kWh (year) per unit of temperature from the area of control, and the maximum and limit value can calculated from the formula below:

$$EP = EP_{H+W} + \Delta EP; \{kWh/(m^2 - year) \tag{1}$$

where

$$EP_{H+W} = \text{the value of EP total for split-type air conditioning, lighting, etc., and} \tag{2}$$

$$\Delta EP_c = \text{the EP value for the air conditioning system;} \tag{3}$$

$$\Delta EP_L = \text{the EP value for the lighting system.} \tag{4}$$

The installation of multi-unit split-type air conditioning in each room does not include the lighting system, which is calculated from the formula below:

$$\Delta EP_c = 10A_{f,c}/A_f; (kWh/m^2. year) \tag{5}$$

where

$A_f$ = the air temperature inside the room considered in the energy performance regulation problem within the building (m$^2$);

$A_{f,c}$ = the air temperature controlled inside the room to determine the regulation (m$^2$) [22] (Figure 2).

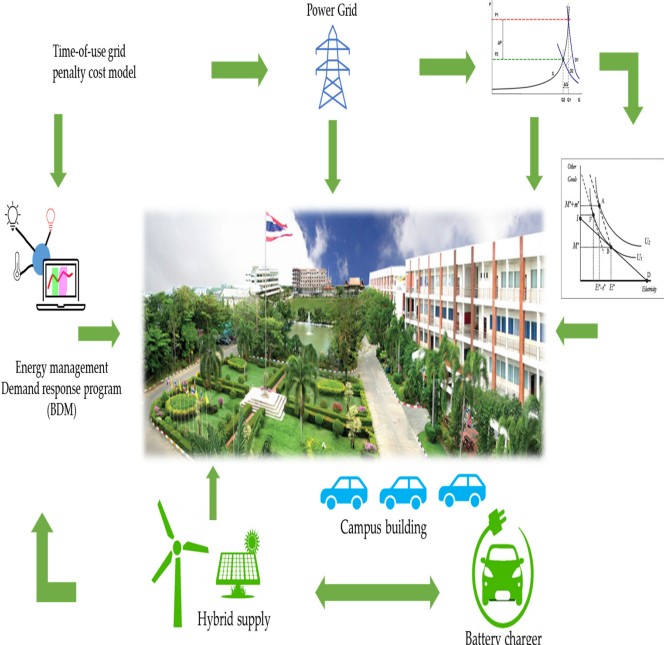

**Figure 2.** Demand response and time response algorithm challenges and issues.

Regarding the involvement of the main electrical circuit control system for the building, it controls the AC and DC power for the loads used and communicates and processes big data using an algorithm between smart sensors and control devices such as smartphones, cloud systems, or tablets. The control electricity from all input units, from all sources such as smart grid operators, passes through an intelligent metering system, or an internal power source (PV, wind turbine, or DG).

The completeness of the power quality system is checked before supplying energy to the main electricity distribution module; then, the request for a current flow load will be fulfilled.

The control function calculates the electric power usage in the building, which is communicated between the control unit and the intelligent sensor, to determine the usage of the power quantity from each device through the Internet gateway module, LAN, Wi-Fi, and a mobile phone (Figure 2).

The smart sensor unit can collect the data and commands from the centralized control unit, measure the electric value, and then send it back to the processor unit. The smart sensor obtains the identification data received through the communication system and commands the equipment to work as a computer. The data are in an encrypted format (Figure 3).

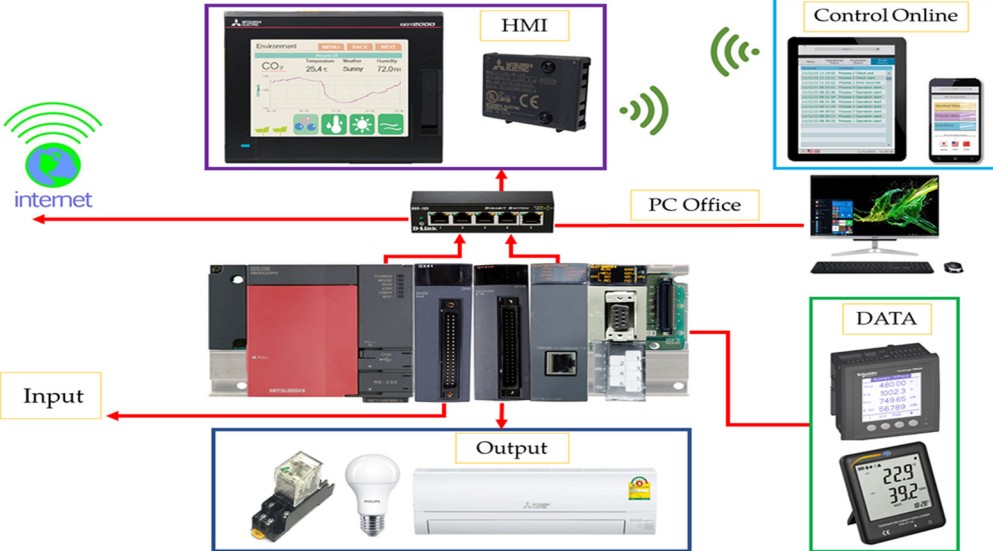

**Figure 3.** Lighting control system for classrooms.

The control algorithm of the lighting system inside each room has control and sends the data via the Internet to each floor, individually processing the data following the algorithm diagram. This includes the central processing for the lighting system and control of its energy use.

The controller unit needs to be aware of the activated device's status and its functioning. The processing unit contains information on the past electricity consumption and the future demand for electricity usage, which can be used to predict the electricity to be used over the next 10 h. The data can be communicated through the radio wave, PLC, wireless, or SCADA system in the building, which controls the receiving and sending of data to the network (cloud system) and produces the identification code for the destination equipment. It is critical in making the system suitable for the required application while all other electrical devices are running (Figure 3).

The processing unit will control the power consumption and manage the power flow in the BEMS algorithm through data communication by restricting the energy used by load devices inside the building, and monitoring and adjusting the power usage of the communication device.

### 3.2. Case Study Data Description

This case study examined the electricity demand of the 20,000-square-meter Thonburi University Engineering Department Building, located in Bangkok, Thailand, a region with hot summers and warm weather. This building has four stories; each story has 10 rooms and includes offices, computer laboratories, teaching classrooms, and many other laboratories within different engineering departments. Each room includes four sets of 38,000 Btu air conditioning units, 36 light bulbs, projector, computer, and other miscellaneous plugs for device accessories (Figure 4).

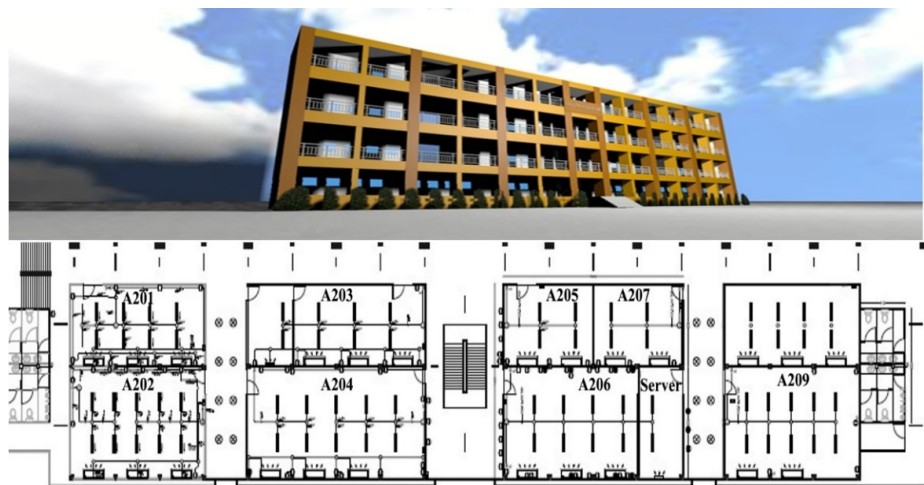

**Figure 4.** Physical model of the building.

The study focused on the demand response strategies by comparing the electric grid electricity costs and the decreased electricity cost impact using a microgrid building management system.

The real-time data for this study were collected using a digital power meter installed in the MDB panel. The digital power meter measures the data and transmits them to the main processor device. The data are sent though the RS485 channel, connecting to the computer where the power management program is installed to study the patterns of electricity use in the engineering buildings (Figure 5).

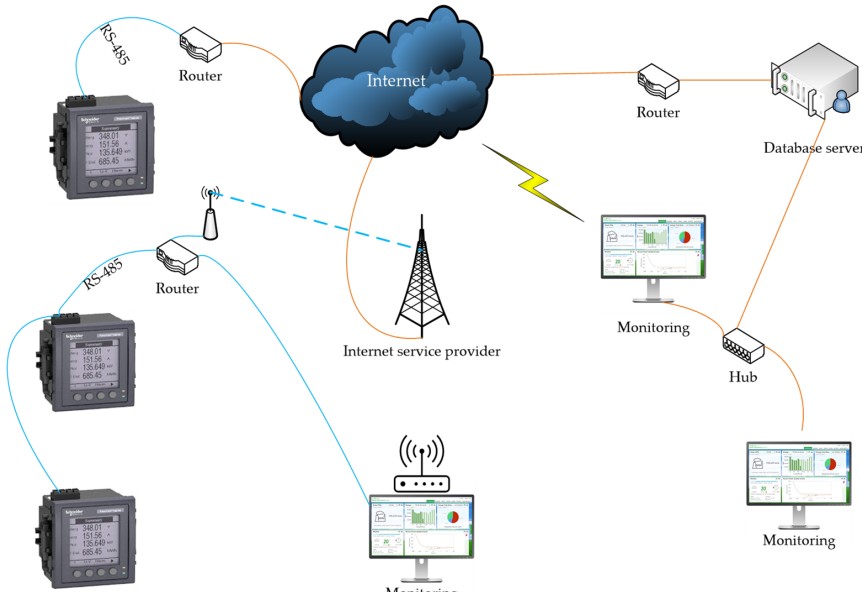

**Figure 5.** Display of the digital power meter.

The electricity consumption, recorded in the computer through the RS485 port, shows the real-time data storage. The electricity consumption can be displayed as both a graphical datasheet and via Microsoft Excel, which shows the total electricity consumed in the MDB for each phase such as phase A, phase B, and phase C for a sample of 30 days (Figure 5).

Demand response configurations in a microgrid building management system can limit each phase of the peak amount of electricity consumed, showing the maximum data every 15 min. The program also shows the average daily electricity consumption and the total electricity consumption in a day, and the data can be viewed in a Microsoft Excel datasheet.

Figure 6 shows the cyclical trend of the three-phase power system for each electrical phase, for regular use without any control. Based on this information, we can observe and generate some warnings, with other options to set up the power quality.

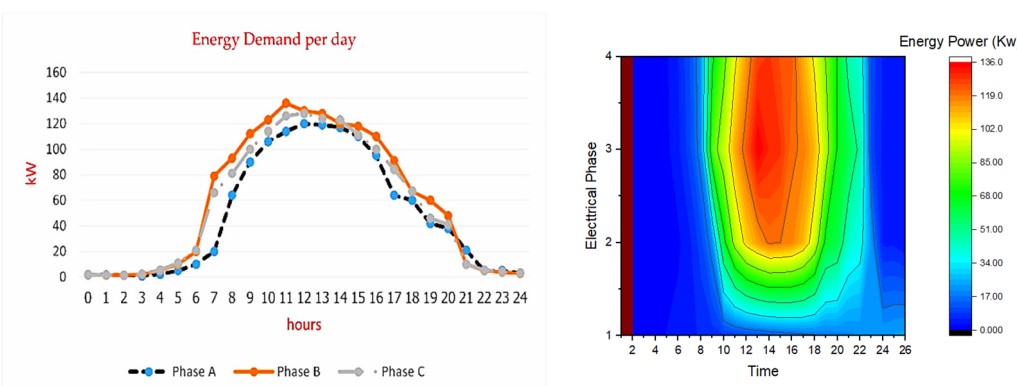

**Figure 6.** Electric grid power consumption without the building demand response.

The combined use of single-phase appliances in a three-phase power system can be connected to a single-phase power system to use different types of electricity, if each powerline is used with different electricity loads, balancing the single-phase and three-phase systems and reducing the electricity usage.

If the power consumption of each line is unbalanced, or the load is very different, the line with more powerlines increases its electricity consumption. In this case, customers pay higher electricity bills at progressive rates.

Figure 6 shows that the load behavior and total electricity power consumption in each phase in 24 h from the demand side were unbalanced and the peak power was 140 kW. Each day, the peak load is different and mainly depends on the classroom schedule occupancy; for example, the weekend class schedule had the most significant electricity consumption.

### 3.3. Temperature Air Condition Control

Air conditioning appliances consume large amounts of energy in Thailand; in the past few years, the government has introduced a campaign to decrease the energy consumption in buildings by controlling the temperature and human behavior to meet comfort conditions. In our case study, the older buildings had mostly installed a split-type of air conditioning system, with many units depending on the office size. The traditional temperature control for air conditioning operates individually, and the indoor temperature was the basis of control for the comfort temperature. The microprocessor becomes essential in controlling the room temperature for all split-type air conditioning systems using an on–off AC compressor. The microprocessor reads the indoor temperature and outdoor temperature via sensors installed inside and outside to compare the temperatures and sends the data to be processed. The advantage of an on–off control temperature system is that it can set up a comfort zone inside the room because it uses more than one temperature sensor per AC unit.

The energy usage from the one-day data monitoring system was compared regarding the total energy for the building and the air conditioning energy usage for the building,

as shown in Figure 7. The energy usage for an air conditioner creates the highest demand for the total energy for the building. The demand response curves are presented, which peaked at nearly 120 kW-hour due to the outside temperatures.

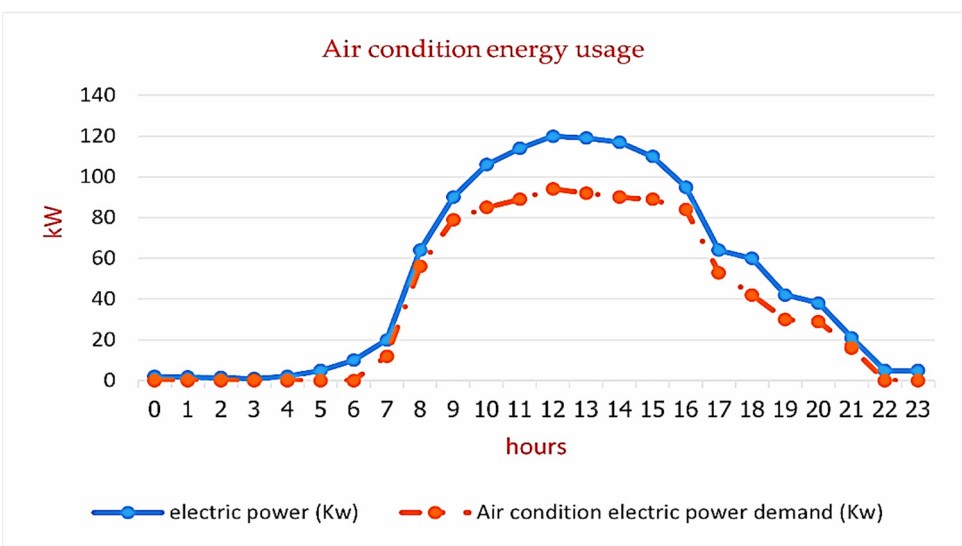

**Figure 7.** Air condition energy usage demand on average in kilowatts.

The transfer function of air conditioning and the room space, sensor, and controller were organized to sample the response of the indoor temperature and compressor output for the AC control algorithm. A traditional air conditioner uses a temperature sensor for feedback. This causes a different control problem, wherein the control output relies on sensing between the setting point and the actual feedback signal. Here, methodologies are presented that use new control methods to prevent the need for more sensors to sense a signal under additional conditions and implement more algorithms to achieve a comfortable temperature.

The split type of air conditioner for each room included three units for each room. The problem of temperature control for air conditioning is that it does not send real-time temperature data to switch the air compressor on or off, which uses more energy power and increases the energy consumption demand every day. The new controller unit presented in Figure 8 uses three units of temperature sensors, DHT22, combining the data and control more accurately with the data following the algorithm.

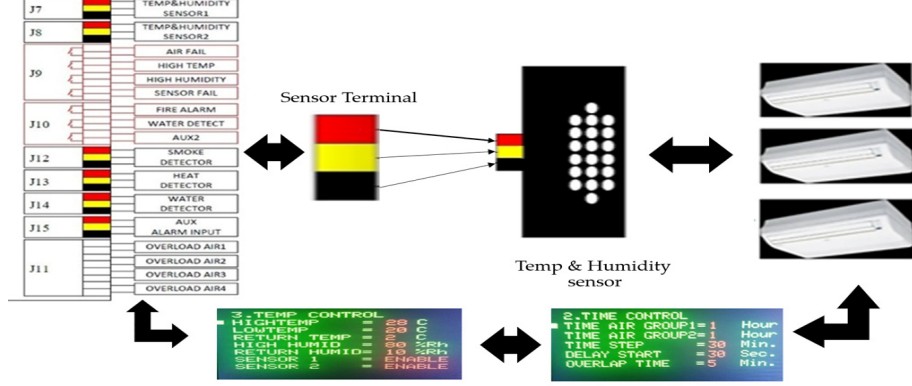

**Figure 8.** The centralized multi-unit air conditioning temperature and humidity control.

Figure 9 shows the diagram of the split-type of air conditioning control unit. The schematic diagram of the split-type AC system process involves the following steps.

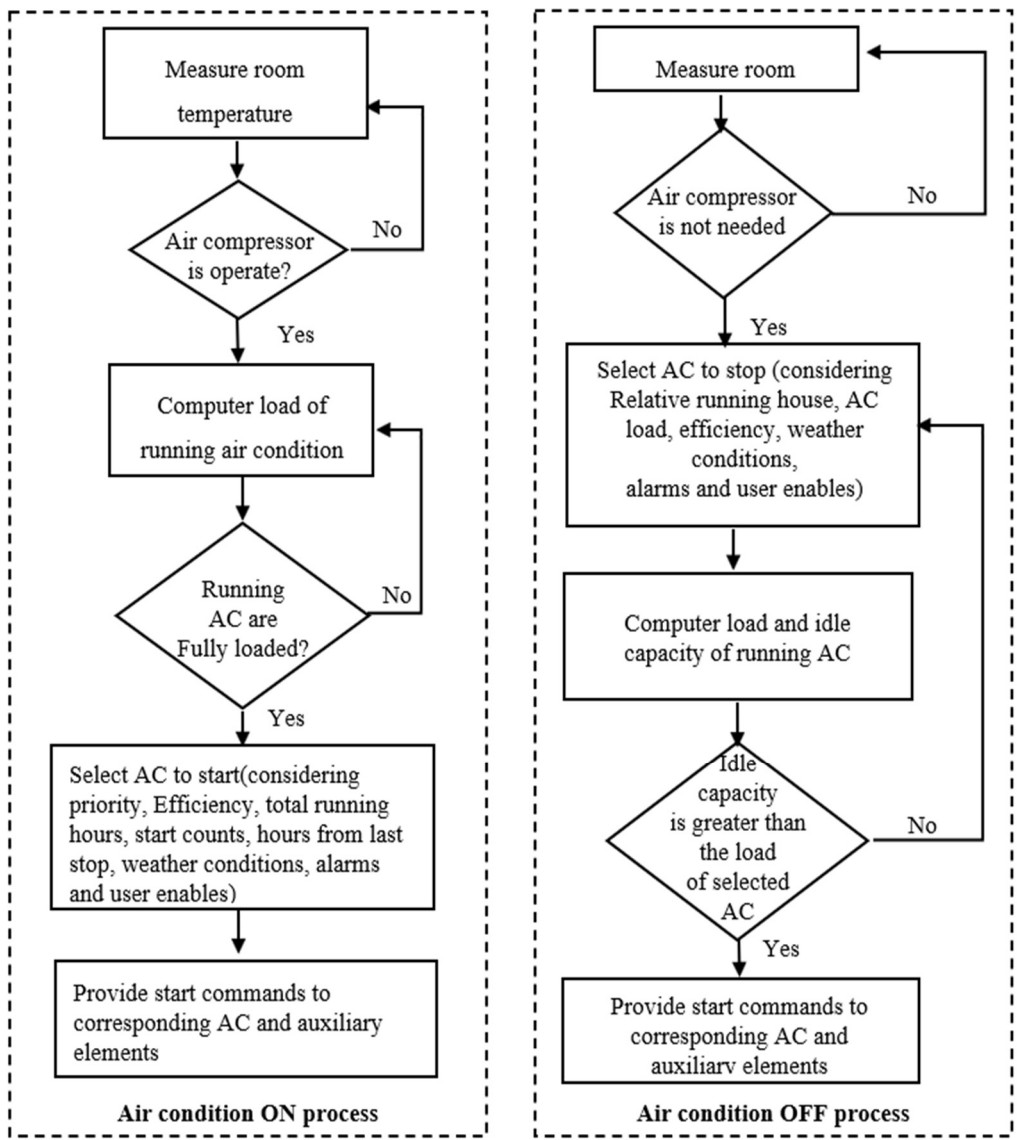

**Figure 9.** Flow chart of the air conditioning temperature ON–OFF process.

Air conditioning ON process:

Step 1: Measure the room temperature to calculate the time for air compressor setup.

Step 2: Air compressor ON or OFF process.

Case 1: ON process is set up for AC compressor running.

Case 2: OFF process is setup for AC fan running only.

Step 3: Computer load command for running air conditioning.

Step 4: Running AC is fully loaded.

Case 1: Preventive AC to start (considering priority, efficiency, total running hours, start count, hours from last stop, weather conditions, alarms, and user activation).

Case 2: Consideration of temperature for running air conditioning.

Step 6: Provide start commands to corresponding AC and auxiliary elements.

Air conditioning OFF process:

Step 1: Measure room temperature.

Step 2: Determine whether air compressor is still needed.

Case 1: No condition process, cooling fan on, continue checking temperature process.

Case 2: Yes, condition process, process AC compressor condition.

Step 3: Select AC to stop (considering relative running house, AC load, efficiency, weather conditions, alarms, and user activation).

Step 4: Computer load and idle capacity of running AC.

Step 5: Idle capacity is more significant than the load of the selected AC.

Case 1: No condition process, running process from step 3.

Case 2: Yes, condition process, consideration of temperature for running air conditioning.

Step 6: Provide start commands to the corresponding AC and auxiliary elements.

*3.4. Illumination Control*

Lighting in the building was applied to various surfaces; when considering the effective levels for eye comfort, it is necessary to include the brightness caused by high contrast. The visual impact of eye contrast can induce an area to create low or high brightness. Generally, the reflectance of the workplace is between 70% and 90%, or more than 30 lux; with the uniform level of more than 0.1, the reflection value from the wall is 50–80%, or illumination not less than 50 lux. The use of lighting energy for 24 h data monitoring was compared between the total energy for the building and the energy consumption of the lighting system for the building. The lighting energy consumption creates the aggregate demand of the building. The demand response curves presented a peak of nearly 10 kW because the brightness in the building must be sufficient, as shown in Figure 10.

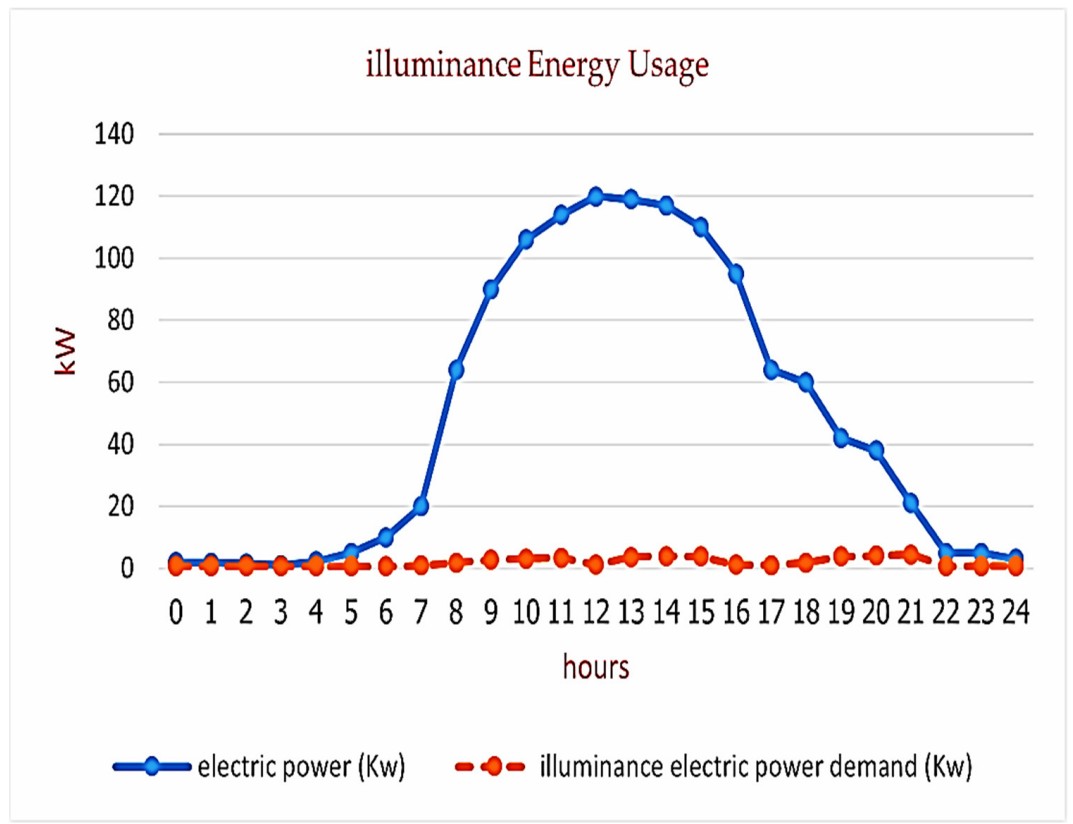

**Figure 10.** Illumination energy usage demand on average in kilowatts.

The technique of lighting control is specific to each room; the main processing control uses PLC to operate data and command data though a soft GOT program to operate the control program. The display system is presented by using the GOT screen via a mobile device through a wireless network connection. The display system can be displayed on the device via a web browser such as IE, Google Chrome, or Firefox, or it can be operated from the touch screen. The PLC can receive the command from the GOT mobile device and the touch screen connects with the output device such as to delay the control of the lighting system in the building, as shown in Figure 11.

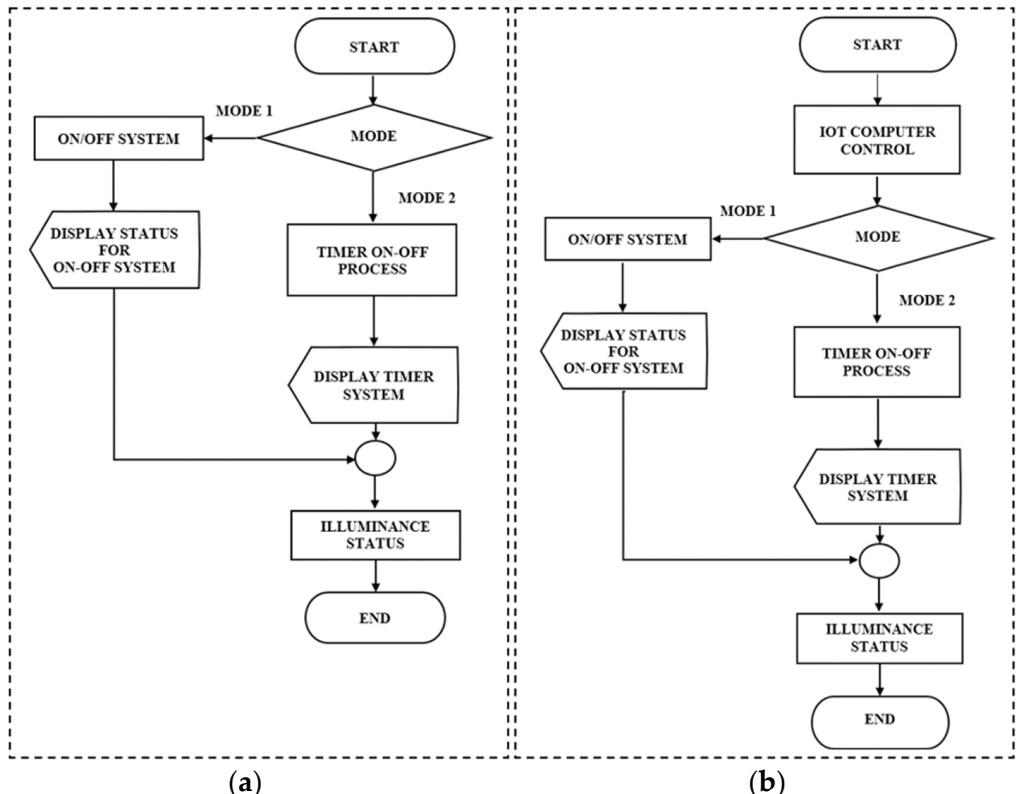

**Figure 11.** Manual and automatic lighting control system: (**a**) manual control; (**b**) automatic control.

In Figure 11, the algorithm for the lighting control via touch screen will control the lighting in front of the control panel, operating to work in two types: it may choose to turn on or off the lighting system manually and set the time to turn on or turn off the light. In type 1, the touch screen will divide the buttons to work with all lighting zones to be used and control the lighting in each zone, where each push button is a function and can be pressed on or pressed off, and it displays the status of each zone.

The concept of remote lighting control via a mobile phone can be used to control the lighting system from a web browser using an Internet signal. The signal is emitted from the control cabinet, and the screen displayed on the phone is the same as the touch screen and can be used in the same way as all touch screens, whether it is manually turned on or off or uses a timer to turn the system on or off.

### 3.5. Case Study Analysis—Results

It is essential to ensure the optimal use of electricity from the electric grid during the electricity peak demand for selected days by using 20 kW of the renewable electric energy installed in the building electricity system. During the electric load peaks, it can be compensated during the weekend classes that take place from 9:00 a.m. to 21:00 p.m., bringing savings regarding the total electricity cost.

Figure 12 shows that the highest electricity consumption occurred in phase A of the electric powerline. To balance all three phases, the demand response and time response algorithm, chosen to decrease the peak electric power by 15%, was used during the weekend classes.

The renewable energy collected into the battery storage device replaces the electricity supply from the electric grid into the building's electric powerline, as shown in Figure 12.

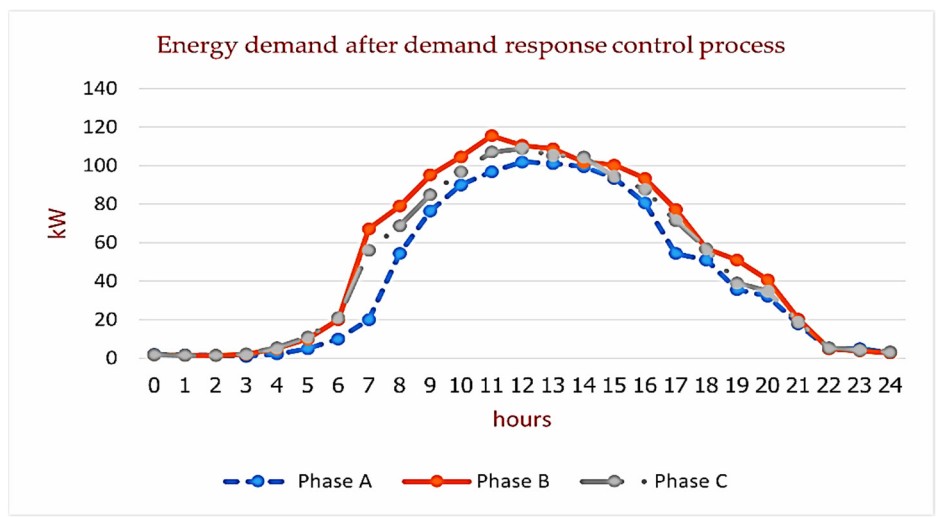

**Figure 12.** Electric grid power consumption after a decrease of 15% due to the building demand response.

## 4. The Implementation of the Building Demand Response (BDR): Issues and Challenges

The problems of implementing the building demand response and time response algorithm are based on several key challenges (Table 1).

**Table 1.** Demonstrates the problems and challenges of the demand response program and requirements to identify limiting problems.

| Issue | Algorithm Challenge | Situation |
| --- | --- | --- |
| Modeling the electricity demand load | Building demand response and time response algorithm analyzes the changes in the electricity consumption behavior pattern and electricity peak demand. The development of the algorithm characteristics includes the electricity demand load and the aggregate demand load. | Develop the electricity load model based on the HVAC historical load demand data including the weather forecast, which reduces or increases the HVAC electricity load demand. |
| Coordination strategy | The building energy management coordination with the smart electric grid and the renewable energy stored as electric energy. | The smart electric grid technology in use with programmable control power to switch on/off the power connection. |
| Hardware and software platform | ICT with different communication types between the sensor devices and the energy management system control. | Building energy management with multiple communication supports a standard base for easy integration. |
| Renewable energy load demand-based power | The demand response and time response of the building algorithm aims to set a comfort zone operational level with energy control as a priority, and operation pattern control of the split-type AC, lighting, and electric appliances including the electricity load demand, the time of electricity used, and the building behavior | The algorithm needs to explore the electricity required as load demand prediction, considering the weather forecast parameters and other setup variables, for the algorithm to find the best solution to the building demand response and time response |
| Warning system | BDR algorithm shows the most cost-effective results and its limitations | The marketing strategies, adaptation, and education learning process |

Table 1 shows the problems and challenges of demand response and response time requirements to identify limiting problems and situations must be considered in the algorithm. The structure of the load model describes the usage patterns and changes in the building usage from the minimum load to the maximum load.

The development of algorithms and models has focused on the parameters for load models to determine the needs of the comfort demand response, and it also considers weather forecasting in a smart building and grid collaboration strategy to control the electricity from renewable energy.

The software and hardware options of the devices that can be used in the communication system are wired or wireless. The power generation model is a real-time system

and includes building demand load data and alarm systems, which showed the most cost-effective results and constraints.

## 5. Conclusions

In this article, we present the real-time intelligent data control of a demand response program for a low-rise university building that takes into consideration the centralization of a multi-unit split-type air conditioning control and illumination system. From the testing of this system, the proposed design obtained real-time intelligent data from the datalogging system on an IOT platform, displaying and recording data on the cloud server temperature control, and the inside and outside temperature control for lower energy consumption, resulting in a total decrease of 15%. The lighting control using a smart sensor obtained a total decrease 10% with an additional 15 kW from 8.00 to 21.00, and the external power source of solar power connected to an inverter fed the power into each phase of the main distribution, operating at 10.00–15.00, to achieve the maximum energy consumption at 12–18 kWh or 8% of the total energy consumption. Additionally, techniques to prove the algorithm's efficiency and its methodology are described; however, further studies of other building constraints to be applied on similar problems need further evaluation and model integration.

In summary, the building demand response, time response, and standard operational method need to be cost-effective for the economical operation of the electric grid. The available renewable energy devices and the various electricity network types can be integrated into the building energy management system in the use of renewable energy devices under an optimal schedule.

As future work, new data from each classroom will be analyzed to apply the algorithm and analyze its performance to approach the implementation of an actual platform to achieve sustainable procedure control. The large PV system needs to be upgraded to generate more power during the peak time, combined with multiple choices of renewable energy sources.

**Author Contributions:** Conceptualization, A.N. and B.P. All authors have read and agreed to the published version of the manuscript.

**Funding:** This research received no external funding.

**Institutional Review Board Statement:** Not applicable.

**Informed Consent Statement:** Not applicable.

**Data Availability Statement:** Not applicable.

**Conflicts of Interest:** The authors declare no conflict of interest.

## Abbreviations

The following abbreviations were used in this manuscript:

| | |
|---|---|
| AC | Alternating current |
| BEMS | Building energy management system |
| BDR | Building demand response |
| DC | Direct current |
| DG | Distribution generator |
| DR | Demand response |
| EP | Electric power |
| HVAC | High-voltage air conditioning |
| ICT | Information and communication technology |
| IOT | Internet of Things |
| kW | Kilowatts |
| PLC | Programable logic control |
| PV | Photovoltaic |
| SCADA | Supervisor control and data acquisition |

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
