# Peer review of "Improving Energy Management through Demand Response Programs for Low-Rise University Buildings"

_sustainability, doi:10.3390/su142114233_

Round 1

Reviewer 1 Report

The article, titled '' The Improvement of Energy management at Private University Building considering on air condition and illumination based on Demand Response Program'' is presented for evaluation. The article is interested, however, it require some changes. Following are the comments for authors to improve the article:

1. Title of the article should be changed, as representing private building give no significance addition. Title should be attractive and scientifically improved. 

2. There should be problem statement / need of article at the start of the abstract.

3. There are so many papers already available in the literature on this topic, authors should revise the abstract such as it should reflect some novelty of the present work.

4. There should be some statistical / figured / numeric values of the results and/or outcome of this study, required for the better and easy read of the viewers of the article.

5. There are only 3 keywords, which should be enhanced to 5 as normal practice of the research papers.

6. The introduction starts with the very general statement, which should not be the case, as this is research paper and not a general theory discussion.

7. More importantly, authors should not use lump sump references, as this is highly discouraged in the scientific reporting. for example, ref. 1-7 are given after 2 line statement. This is very general statement, authors should be aware of the fact that where a reference is required, there should be specific information. In general practice, no more than two references are used in any statement.

8. Only the first para of the introduction , which is very general, is about 23 references, which should not be the case and authors should revise the intro section accordingly.

 9.  The equations used are from the literature? If yes, please cite properly.

10. Figure 3 is simple figure of building, which should not be part of the research paper. It add no significant value in the article. Please maintain the research paper rythm. 

11. Figure 4 is highly discouraged, as these are screenshots of the software. Authors should draw proper results instead of adding raw results / methodology in the research paper. The display of display meters must not be the part of any research paper.

12. Figure 5 contain no units. how authors can justify this data? 

13. Again figure 6 contain no units on y-axis. Authors have not properly reviewed the article before submission.

14. Figure 7 is of very poor quality and authors have drag the figure improper adjustment and it become non-visible with respect to text in the figure. 

15. Figure 8 is the general process flow diagram and is available in the literature . It should be cited accordingly, if taken from literature.

16. Again figure 9 has no y-axis units and authors must redraw all figures again with high quality and to do some data analyses of the results. These figures are mere a repetition and seems not interesting without any proper data analyses. 

17. Figure 10 must be somewhere in the methodology section.  

18. same is the case with figure 12.

19. Ins section 4, table is there but without any caption or heading etc. 

20. In the text, there is ''table 1'', but there is no Table 1 heading. authors must proof read the article before submission. 

21. English language structure is very poor, minor typos etc are there for correction.

22. Conclusion section should be re-written, as it present only a general information. 

23. There are many many old references. In the recent past, there is a lot of work is carried out on this topic. authors should make sure that no older references (3-5 years) should be in the paper. 

Reviewer 2 Report

-The English Language is fine, yet  general check is recommended.
-Additional suggestions to improve the paper, in which I make reference to the second column of line numbers, are reported below:
-the performance ratio should be defined in a few word in the abstract as it is not a generally accepted performance parameters.

Reviewer 3 Report

The paper titled “The Improvement of Energy management at Private University Building considering on air condition and illumination based on Demand Response Program” is a well-organized paper. Before publication, the following items should be considered by the author:

1.       English must be improved.

2.       The novelty of the paper should be highlighted.

3.       The most important quantitative results must be added to the abstract.

4.       The Journal’s standards for referencing must be considered.

5.       A list of abbreviations and acronyms should be prepared.

6.       Avoid attributing several references to a simple sentence (for instance, in the introduction section, seven references have been cited in just one simple sentence).

7.       The literature review and problem statement should be developed.

8.       The following papers are suggested to be considered:

“Theoretical Analysis of the Performance and Optimization of Indirect Flat Evaporative Coolers”. Future Energy (2022), 2 (1):9-14. DOI: 10.55670/fpll.fuen.2.1.2

"An overview of the building energy management system considering the demand response programs, smart strategies and smart grid." Energies 13.13 (2020): 3299.

9.       All equations need an appropriate reference.

10.   A list of acronyms must be prepared.

11.   The quality of Fig4 is not good.

Round 2

Reviewer 1 Report

Accept

Reviewer 3 Report

Accept